# Is PRM Necessary? Problem-Solving RL Implicitly Induces PRM Capability in LLMs

**Zhangyin Feng**[*]
Huawei Technologies Ltd.
fengzhangyin@huawei.com

**Qianglong Chen**[*]
Huawei Technologies Ltd.
chenqianglong.ai@gmail.com

**Ning Lu**
HKUST
nluab@cse.ust.hk

**Yongqian Li**
Huawei Technologies Ltd.
liyongqian2@huawei.com

**Siqi Cheng**
Huawei Technologies Ltd.
chengsiqi4@huawei.com

**Shuangmu Peng**
Huawei Technologies Ltd.
pengshuangmu@huawei.com

**Duyu Tang**
Huawei Technologies Ltd.
tangduyu@huawei.com

**Shengcai Liu**[†]
Guangdong Provincial Key Laboratory of Brain-Inspired
Intelligent Computation, Department of CSE, SUSTech
liusc3@sustech.edu.cn

**Zhirui Zhang**[†]
Huawei Technologies Ltd.
zrustc11@gmail.com

## Abstract

The development of reasoning capabilities represents a critical frontier in large language models (LLMs) research, where reinforcement learning (RL) and process reward models (PRMs) have emerged as predominant methodological frameworks. Contrary to conventional wisdom, empirical evidence from DeepSeek-R1 demonstrates that pure RL training focused on mathematical problem-solving can progressively enhance reasoning abilities without PRM integration, challenging the perceived necessity of process supervision. In this study, we conduct a systematic investigation of the relationship between RL training and PRM capabilities. Our findings demonstrate that problem-solving proficiency and process supervision capabilities represent complementary dimensions of reasoning that co-evolve synergistically during pure RL training. In particular, current PRMs underperform simple baselines like majority voting when applied to state-of-the-art models such as DeepSeek-R1 and QwQ-32B. To address this limitation, we propose Self-PRM, an introspective framework in which models autonomously evaluate and rerank their generated solutions through self-reward mechanisms. Although Self-PRM consistently improves the accuracy of the benchmark (particularly with larger sample sizes), analysis exposes persistent challenges: The approach exhibits low precision (<10%) on difficult problems, frequently misclassifying flawed solutions as valid. These analyses underscore the need for continued RL scaling to improve reward alignment and introspective accuracy. Overall, our findings suggest that PRM may not be essential for enhancing complex reasoning, as pure RL not only improves problem-solving skills but also inherently fosters robust PRM capabilities. We hope these findings provide actionable insights for building more reliable and self-aware complex reasoning models.

---

[*]Equal contribution.
[†]Corresponding author.

39th Conference on Neural Information Processing Systems (NeurIPS 2025).

# 1 Introduction

Recent advances in reinforcement learning (RL) scaling have substantially improved the ability of large language models (LLMs) to perform complex reasoning tasks. This progress is exemplified by the emergence of state-of-the-art (SOTA) models such as Kimi 1.5 [25], OpenReasonerZero [10], DeepSeek-R1 [8], QwQ-32B [32, 26], and Claude-3.7-Sonnet [1], all of which demonstrate remarkable problem-solving capabilities achieved through pure RL training.

Alongside this RL-driven progress, process reward models (PRMs) [16, 30, 2] have emerged as an alternative paradigm for reasoning supervision, with the aim of evaluating and optimizing the quality of intermediate reasoning steps. Despite their conceptual appeal, PRMs encounter three fundamental limitations: (1) the inherent ambiguity in defining granular reasoning steps, (2) prohibitive annotation costs, and (3) systemic vulnerability to reward hacking mechanisms. In particular, the DeepSeek-R1 technical report highlights these limitations [8], demonstrating that PRMs did not yield significant gains in the performance of complex reasoning. However, recent work continues to explore the potential of PRMs. For example, PRIME [4] introduces implicit reward mechanisms to bypass the need for labeled steps, GenPRM [37] uses code verification and relative progress estimation to improve PRM performance, and Qwen PROCESSBENCH [38] offers a benchmark to assess the quality of the reasoning trace. These efforts underscore the ongoing interest in PRMs as a path to better reasoning supervision. However, a crucial question remains largely unexplored: *Is explicit PRM training truly necessary for developing models capable of process-level reasoning?* In other words, can RL training alone – in the absence of any process supervision – lead to models that are both strong problem solvers and capable evaluators of reasoning quality?

To answer this, in this work, we present the first systematic empirical study exploring the intrinsic connection between RL training and PRM capabilities. Our key insight is that problem-solving and process supervision are not orthogonal skills, but rather complementary competencies that co-evolve during pure RL training. Through a series of experiments, we examine how reasoning models trained solely with rule-based rewards develop strong process-level judgement capabilities, without access to fine-grained supervision. Through a series of controlled experiments on math reasoning tasks, we demonstrate that RL-trained models like DeepSeek-R1 and QwQ-32B exhibit strong process judgement abilities, exceeding those of models explicitly trained with PRMs.

In addition, we find that existing PRMs provide little to no benefit when applied to these models for reranking, often underperforming compared to simple heuristics such as majority voting. To address this, we introduce Self-PRM, an introspective approach where a model leverages its own internal reward signal to re-rank sampled outputs. Self-PRM consistently improves performance – particularly at higher sample sizes – by aligning candidate selection with the model's own reasoning preferences. Despite these improvements, a detailed analysis reveals that Self-PRM suffers from low precision on challenging problems, often misclassifying incorrect solutions as correct. Stronger models, such as DeepSeek-R1 and QwQ-32B, exhibit more reliable self-evaluation, but the issue persists, highlighting the limitations of introspective scoring when reward alignment is imperfect. These findings challenge conventional assumptions about the necessity of process supervision and suggest that lightweight, introspective training signals – whether through self-evaluation or continued RL scaling – may provide a viable path forward.

In summary, our main contributions are as follows:

- We provide the first systematic study demonstrating that RL training alone – without any process-level supervision – can endow models with strong PRM capabilities.

- We empirically show that existing PRMs fail to improve performance when applied to strong RL-trained models, and often underperform simple baselines such as majority voting.

- We introduce the Self-PRM framework, in which models rerank their own outputs using internal reward signals, achieving consistent performance gains – while also revealing limitations in precision that highlight the need for improved reward alignment.

# 2 Preliminaries

**Reinforcement Learning.** RL provides a framework in which an agent learns to make sequential decisions by interacting with an environment to maximize cumulative rewards. Formally, RL is

Table 1: Evaluation results of different LLMs on PROCESSBENCH. We report the error, correct and F1 score of the respective accuracies on erroneous and correct samples.

| Model | GSM8K | | | MATH | | | OlympiadBench | | | OmniMATH | | | Average |
|---|---|---|---|---|---|---|---|---|---|---|---|---|---|
| | Error | Correct | F1 | Error | Correct | F1 | Error | Correct | F1 | Error | Correct | F1 | |
| *Proprietary LLMs (Critic)* | | | | | | | | | | | | | |
| GPT-4o-0806 | 70.0 | 91.2 | 79.2 | 54.4 | 76.6 | 63.6 | 45.8 | 58.4 | 51.4 | 45.2 | 65.6 | 53.5 | 61.9 |
| o1-mini | 88.9 | 97.9 | 93.2 | 83.5 | 95.1 | 88.9 | 80.2 | 95.6 | 87.2 | 74.8 | 91.7 | 82.4 | 87.9 |
| *DeepSeek R1 and R1 Distilled Models, prompted as **Generative PRM*** | | | | | | | | | | | | | |
| R1-Distill-Qwen-7B | 45.9 | 90.2 | 60.8 | 51.5 | 82.8 | 63.5 | 35.6 | 73.5 | 47.9 | 27.8 | 70.1 | 39.8 | 53.0 |
| R1-Distill-Qwen-32B | 74.4 | 98.4 | 84.7 | 71.5 | 89.4 | 79.5 | 64.3 | 85.5 | 73.4 | 59.0 | 83.8 | 69.3 | 76.7 |
| DeepSeek-R1 | 84.1 | 95.3 | 89.3 | 82.3 | 91.1 | 86.5 | 78.2 | 86.4 | 82.1 | 71.7 | 80.9 | 76.0 | 83.5 |
| *Qwen-series Models, prompted as **Generative PRM*** | | | | | | | | | | | | | |
| Qwen2.5-Math-7B-Instruct | 15.5 | 100.0 | 26.8 | 14.8 | 96.8 | 25.7 | 7.7 | 91.7 | 14.2 | 6.9 | 88.0 | 12.7 | 19.9 |
| Qwen2.5-Math-72B-Instruct | 49.8 | 96.9 | 65.8 | 36.0 | 94.3 | 52.1 | 19.5 | 97.3 | 32.5 | 19.0 | 96.3 | 31.7 | 45.5 |
| Qwen2.5-7B-Instruct | 40.6 | 33.2 | 36.5 | 30.8 | 45.1 | 36.6 | 26.5 | 33.9 | 29.7 | 26.2 | 28.6 | 27.4 | 32.6 |
| Qwen2.5-14B-Instruct | 54.6 | 94.8 | 69.3 | 38.4 | 87.4 | 53.3 | 31.5 | 78.8 | 45.0 | 28.3 | 76.3 | 41.3 | 52.2 |
| Qwen2.5-32B-Instruct | 49.3 | 97.9 | 65.6 | 36.7 | 95.8 | 53.1 | 25.3 | 95.9 | 40.0 | 24.1 | 92.5 | 38.3 | 49.3 |
| Qwen2.5-72B-Instruct | 62.8 | 96.9 | 76.2 | 46.3 | 93.1 | 61.8 | 38.7 | 92.6 | 54.6 | 36.6 | 90.9 | 52.2 | 61.2 |
| QwQ-32B | 84.1 | 97.4 | 90.2 | 83.0 | 90.4 | 86.5 | 75.9 | 87.3 | 81.2 | 71.1 | 83.8 | 77.0 | 83.7 |
| *Other Models with Training PRM data, as **Discriminative PRM*** | | | | | | | | | | | | | |
| Skywork-PRM-1.5B | 50.2 | 71.5 | 59.0 | 37.9 | 65.2 | 48.0 | 15.4 | 26.0 | 19.3 | 13.6 | 32.8 | 19.2 | 36.4 |
| Math-Shepherd-PRM-7B | 32.4 | 91.7 | 47.9 | 18.0 | 82.0 | 29.5 | 15.0 | 71.1 | 24.8 | 14.2 | 73.0 | 23.8 | 31.5 |
| RLHFlow-PRM-Mistral-8B | 33.8 | 99.0 | 50.4 | 21.7 | 72.2 | 33.4 | 8.2 | 43.1 | 13.8 | 9.6 | 45.2 | 15.8 | 28.4 |
| RLHFlow-PRM-Deepseek-8B | 24.2 | 98.4 | 38.8 | 21.4 | 80.0 | 33.8 | 10.1 | 51.0 | 16.9 | 10.9 | 51.9 | 16.9 | 26.6 |
| Skywork-PRM-7B | 61.8 | 82.9 | 70.8 | 43.8 | 62.2 | 53.6 | 17.9 | 31.9 | 22.9 | 14.0 | 41.9 | 21.0 | 42.1 |
| Qwen2.5-Math-7B-Math-Shepherd | 46.4 | 95.9 | 62.5 | 18.9 | 96.6 | 31.6 | 7.4 | 93.8 | 13.7 | 4.0 | 95.0 | 7.7 | 28.9 |
| Qwen2.5-Math-7B-PRM800K | 53.1 | 95.3 | 68.2 | 48.0 | 90.1 | 62.6 | 35.7 | 87.3 | 50.7 | 29.8 | 86.1 | 44.3 | 56.5 |
| RetrievalPRM-7B | 64.7 | 88.1 | 74.6 | 67.2 | 75.6 | 71.1 | 56.0 | 65.2 | 60.2 | 52.8 | 62.7 | 57.3 | 65.8 |
| Qwen2.5-Math-PRM-7B | 72.0 | 96.4 | 82.4 | 68.0 | 90.4 | 77.6 | 55.7 | 85.5 | 67.5 | 55.2 | 83.0 | 66.3 | 73.5 |
| Qwen2.5-Math-PRM-72B | 78.7 | 97.9 | 87.3 | 74.2 | 88.2 | 80.6 | 67.9 | 82.0 | 74.3 | 64.8 | 78.8 | 71.1 | 78.3 |

typically modeled as a Markov Decision Process (MDP), defined by the tuple $(\mathcal{S}, \mathcal{A}, \mathcal{P}, \mathcal{R}, \gamma)$, where $\mathcal{S}$ is the state space, $\mathcal{A}$ the action space, $\mathcal{P}$ the state transition dynamics, $\mathcal{R}$ the reward function, and $\gamma$ the discount factor. In the context of LLMs, RL is widely used to align model behavior with human preferences. One prominent approach is Reinforcement Learning from Human Feedback (RLHF) [20], where a reward model (RM) is trained to approximate human judgements over generated outputs. The base language model is then optimized using RL to maximize the reward provided by this RM. Several methods have been proposed to implement this optimization process effectively, including PPO [22], DPO [21], GRPO [23], DAPO [35] and VAPO [36] etc. These methods differ in how explicitly they use reward signals, but all share the goal of training LLMs to produce outputs that are more helpful, truthful, and aligned. Notably, recent RL-based models – including DeepSeek-R1 [8], Kimi1.5 [25], QwQ-32B [26] , OpenReasonerZero [10] – achieve state-of-the-art performance on complex reasoning tasks without process-level supervision. Meanwhile, some works [7, 12, 24, 14, 5] are exploring RL with tools for efficient reasoning. While Retool [5] use DAPO-MATH-17k and DAPO for RL training to guide LLMs towards optimal strategies for leveraging external computational tools during reasoning, ToRL [14] leverage 7B-base to improve the complex reasoning ability.

**Process Reward Models.** PRMs [2, 15, 16, 17, 29, 30, 39, 27, 13] are designed to evaluate the reasoning process of LLMs, not just the final answer. They provide rewards or scores for each intermediate step in a reasoning path, helping models learn to think in a more structured, logical, and interpretable way. While PRM800K [15] is proposed to train the reward model, AlphaMath [2] leverage Monte Carlo Tree Search (MCTS) to unleash the potential of a well-pretrained LLM to autonomously enhance its mathematical reasoning. PRM is especially useful for tasks that require multi-step problem solving, such as math or logical reasoning. Compared to traditional outcome-supervised reward models [3, 34] that focus only on whether the final answer is correct, PRMs can give more detailed feedback, making them valuable for improving model alignment and transparency. They are typically trained on labeled reasoning examples, learning to distinguish good reasoning steps from incorrect or irrelevant ones. Despite their potential, PRMs face challenges such as limited high-quality reasoning data, difficulties in defining stepwise correctness, and risks of models exploiting the reward function. To improve PRMs, researchers are exploring better data collection methods, combining them with reinforcement learning algorithms (e.g., PPO [22]), and even letting models evaluate their own reasoning to enhance learning [2] or extend them to visual reasoning [29].

Table 2: Chi-square test results for model performance on PROCESSBENCH, including GSM8K, MATH, OlympiadBench, and OmniMath datasets. We classify the math problems into `True` and `False` categories, depending on whether the LLMs can provide correct solutions. Similarly, we categorize the judgment results into `Correct` and `Error` categories, based on whether the LLMs deliver the correct judgments.

| Model | Solution | GSM8K | | | MATH | | | OLYMPIADBENCH | | | OMNIMATH | | | ALL | | |
|---|---|---|---|---|---|---|---|---|---|---|---|---|---|---|---|---|
| | | Correct | Error | $p$ | Correct | Error | $p$ | Correct | Error | $p$ | Correct | Error | $p$ | Correct | Error | $p$ |
| Qwen2.5-32B | True | 327 | 52 | 2.9e-5 | 599 | 201 | 0 | 298 | 131 | 0 | 243 | 161 | 0 | 1467 | 545 | 0 |
| | False | 11 | 10 | | 63 | 137 | | 220 | 351 | | 134 | 462 | | 428 | 960 | |
| R1-Distill-Qwen-32B | True | 315 | 47 | 0.071 | 755 | 159 | 3.08e-3 | 439 | 146 | 3.21e-3 | 444 | 171 | 0 | 1953 | 559 | 0 |
| | False | 29 | 9 | | 33 | 17 | | 276 | 139 | | 206 | 179 | | 544 | 344 | |
| QwQ-32B | True | 338 | 42 | 0.011 | 826 | 147 | 0.125 | 507 | 110 | 0.023 | 543 | 149 | 3.9e-5 | 2214 | 448 | 0 |
| | False | 14 | 6 | | 20 | 7 | | 292 | 91 | | 204 | 104 | | 530 | 208 | |
| DeepSeek-R1 | True | 330 | 44 | 0.025 | 831 | 141 | 0.039 | 476 | 122 | 1.15e-3 | 501 | 167 | 0.012 | 2138 | 474 | 0 |
| | False | 19 | 7 | | 20 | 8 | | 284 | 118 | | 224 | 108 | | 547 | 241 | |

# 3 Intrinsic Connection between RL Training and PRM

## 3.1 Experimental Setup

We evaluate the PRM capabilities of various models using PROCESSBENCH, a publicly available benchmark designed to assess the reasoning abilities of LLMs across diverse domains of mathematical problem-solving. PROCESSBENCH comprises four distinct datasets: GSM8K [3], MATH, OLYMPIADBENCH [9], and OMNIMATH [6]. For each dataset, we report three metrics: Error Rate, Correct Rate, and F1 Score. We categorize the evaluated models into three main groups:

- Proprietary LLMs (Critic Models): This group includes closed-source models, such as GPT-4o-0806 [11] and o1-mini [19], which serve as reference points.
- Qwen-series and DeepSeek-series Models (Prompted as Generative PRM): These models are not directly fine-tuned on RM or PRM datasets; instead, they are prompted as PRMs. Representative models include Qwen2.5-Math-7B-Instruct [33] and DeepSeek-R1 [8].
- Models Fine-Tuned with PRM Data (Discriminative PRM): These models are explicitly trained on datasets constructed with PRM-style supervision. Models include Math-Shepherd-PRM [28], Skywork-PRM-7B [18], RetrievalPRM-7B [40], and RLHFlow-PRM-Mistral-8B [31].

Additionally, to further validate the emergence of PRM capabilities during RL training, we conduct experiments using Qwen2.5-7B-Base with DAPO-Math-17k [35], employing DAPO as the policy optimization algorithm. The hyperparameters are set as follows: learning rate of 1e-6, batch size of 256, prompt length of 2048, output length of 10240, group size of 16, clipping ratio (high) of 0.28, overlong buffer length of 4096, and an overlong penalty factor of 1.0.

## 3.2 Empirical Analysis on PROCESSBENCH

**Evaluation Results on PROCESSBENCH.** The evaluation results of different LLMs on PROCESSBENCH are presented in Table 1, which summarizes the PRM performance of various models across the GSM8K, MATH, OLYMPIADBENCH, and OMNIMATH datasets. Our findings indicate that models trained purely with reinforcement learning (RL), such as DeepSeek-R1 and QwQ-32B, exhibit strong inherent PRM capabilities, despite not being explicitly supervised with PRM-labeled data. These RL-trained models consistently outperform discriminative PRM models that were fine-tuned directly on PRM-annotated datasets. For instance, DeepSeek-R1 achieves an average F1 score of 83.5, while QwQ-32B achieves 83.7, both surpassing all models trained with PRM labels in a discriminative setting, including Skywork-PRM-7B (42.1), Qwen2.5-Math-PRM-7B (73.5), and Qwen2.5-Math-PRM-72B (78.3). This suggests that PRM capabilities can emerge implicitly through RL, without the need for explicit PRM supervision.

**Problem-Solving V.S. Judgment: A Statistical Perspective.** We further conduct chi-square tests to evaluate the correlation between problem-solving proficiency and process supervision capabilities in the same model. Specifically, we categorize the mathematical problems in `ProcessBench` into two groups for each model: (1) `True`: Problems for which the LLM generates correct solutions; (2) `False`: Problems where the LLM fails to produce correct solutions. Similarly, we classify the model's

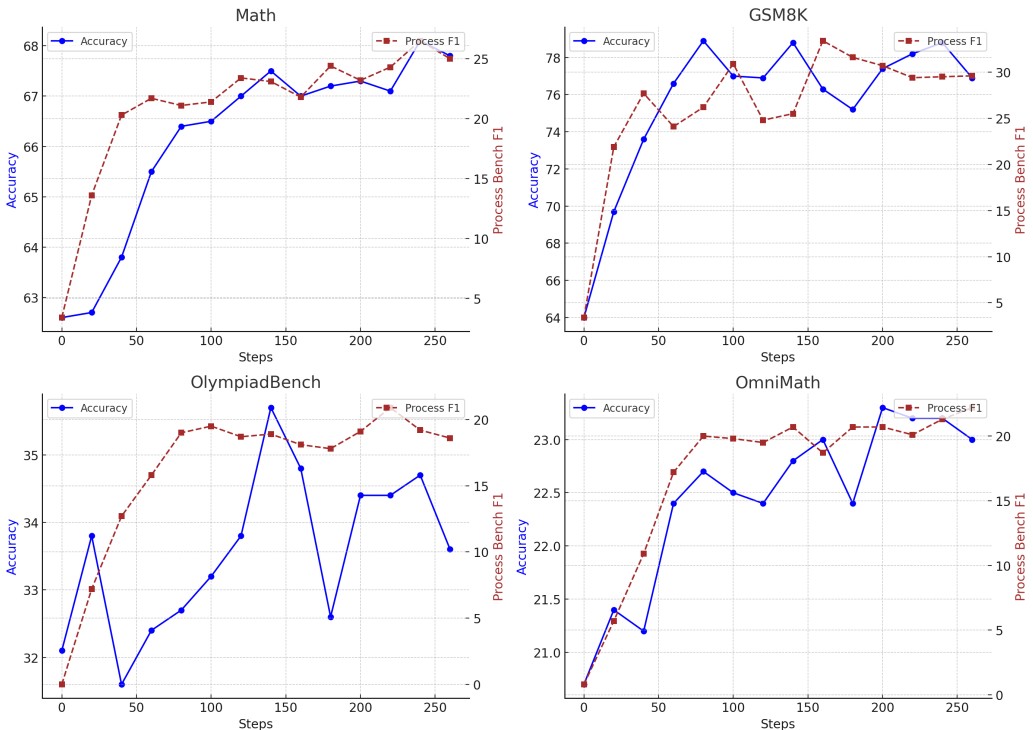

Figure 1: Problem Solving Accuracy and F1 Results of the RL-trained model on ProcessBench. We leverage Qwen2.5-7b-base for RL training.

judgment outcomes into: (1) `Correct`: Cases where the LLM accurately evaluates the correctness of the paired problem-solution instances in `ProcessBench`; (2) `Error`: Cases where the LLM's judgment is incorrect. Based on these categorizations, we count the total number of instances in each of the four groups and calculate the p-value to test the hypothesis that problem-solving proficiency and process supervision capabilities are independent of each other. As summarized in Table 2, all models exhibit statistically significant correlations ($p < 0.05$) in the aggregate analysis, strongly rejecting the null hypothesis. This indicates a strong correlation between problem-solving proficiency and process supervision capabilities. Specifically, for the GSM8K dataset, three models achieve significant results: Qwen2.5-32B ($p = 2.9 \times 10^{-5}$), QwQ-32B ($p = 0.011$), and DeepSeek-R1 ($p = 0.025$), whereas R1-Distill-Qwen-32B did not reach statistical significance ($p = 0.071$). For the MATH dataset, all models except QwQ-32B exhibited significance: Qwen2.5-32B ($p = 0$), R1-Distill-Qwen-32B ($p = 3.08 \times 10^{-3}$), and DeepSeek-R1 ($p = 0.039$). QwQ-32B's performance is non-significant ($p = 0.125$). For the OLYMPIADBENCH and OMNIMATH datasets, all models achieve statistically significant results. The chi-square tests conclusively validate that problem-solving proficiency and process supervision capabilities are intrinsically linked in LLMs, and this correlation persists across diverse mathematical reasoning tasks.

> **Takeaway 1:**
>
> Empirical results demonstrate that RL-trained models (e.g., DeepSeek-R1 and QwQ-32B) develop robust PRM capabilities without explicit PRM supervision, consistently surpassing discriminative PRM baselines. Chi-square analysis ($p < 0.05$) confirms a statistically significant positive correlation between process-level judgment accuracy and problem-solving proficiency.

### 3.3 Can RL Training Improve PRM Capability?

Since the RL-trained models shows strong PRM capability, we further conduct the experiment to observe the change of PRM capability during RL training. As illustrated in Figure 1, we analyze the learning curves of the RL-trained model through four mathematical reasoning benchmarks (GSM8K,

MATH, OLYMPIADBENCH, and OMNIMATH), evaluating both final-answer accuracy (solid blue lines) and ProcessBench F1 scores (red dashed lines). The results demonstrate a consistent and parallel enhancement in both final-answer accuracy and process judgement ability as training steps increase. A salient trend is that gains in process judgement often precede or surpass those in accuracy, particularly in the early training phases. For instance, on the MATH and GSM8K benchmarks, the F1 score on ProcessBench increases sharply within the first 80 steps, while accuracy improves more gradually. This suggests that RL enables the model to internalize and identify valid reasoning structures before fully mastering the corresponding solution strategies. In more challenging problems, such as OLYMPIADBENCH, the F1 score on ProcessBench improves consistently, whereas accuracy exhibits greater volatility. This divergence highlights the value of process-level learning as a stable and transferable signal, even when final-answer correctness is harder to optimize. Similarly, in OMNIMATH, where domain generalization is required, the model achieves modest but consistent improvements in both metrics, indicating the scalability of process supervision under domain shift. These findings indicate that problem-solving and process supervision are not orthogonal skills, but rather complementary competencies that co-evolve during pure RL training.

> **Takeaway 2:**
>
> RL improves both the accuracy of problem solving and the PRM capability in a coordinated manner. Crucially, improvements in process judgement often emerge earlier and more steadily than accuracy, highlighting RL's capacity to implicitly foster interpretable reasoning even in complex or out-of-domain tasks.

## 4 Explore the Performance Limit of Reasoning Models

### 4.1 Experimental Setup

In order to further explore whether a model's own problem-solving ability can enhance its capability as a PRM, we propose the self-reference (Self-REF) paradigm, where the model's generated solutions are used as reference signals to supervise PRM alignment. Using Qwen2.5-32B-Instruct, R1-Distill-Qwen-32B, QwQ-32B, and DeepSeek-R1 as the base model, we evaluate two variants on the PROCESSBENCH benchmark (§4.2): (1) we prompt these models as generative PRMs, and (2) a Self-REF-enhanced PRM incorporating the model's own solutions as supervisory signals.

Meanwhile, we evaluate whether PRMs can enhance the performance of strong reasoning models using two experimental settings (§4.3): (1) BoN w/ PRM, where we use a Qwen2.5-MATH-PRM-72B to select the best output from $k = 8, 16, 32, 64$. (2) BoN w/ Self-PRM, where each model (QwQ-32B, DeepSeek-R1) reranks its own outputs using internally derived reward scores. Evaluations are conducted on AIME24, AIME25, and CNMO24, with performance compared across direct sampling (Pass@k), majority voting, BoN with external PRM, and BoN with Self-PRM. To analyze the limitations of Self-PRM, we further compute the number of solutions labeled correct by Self-PRM ($S_{\text{PRM}}$) and the subset that are truly correct ($S_{\text{TP}}$), allowing us to quantify precision across individual problem indices.

### 4.2 Improving PRM Capability with Self-REF

To understand when self-generated reasoning traces serve as effective supervisory signals for PRM alignment, we access the impact of Self-REF across models with distinct training strategies as shown in Table 3. We observe a clear trend: Self-REF substantially improves PRM performance only for models that have not undergone RL. Specifically, the F1 score of Qwen2.5-32B-Instruct, a purely instruction-tuned model, improves significantly from $45.1 \rightarrow 63.9$ (+18.8) with Self-REF. This gain is most pronounced on MATH (+15.6 F1) and OlympiadBench (+13.9 F1), indicating that Self-REF can inject meaningful structure into process modeling where it is otherwise absent. In contrast, RL-trained models such as QwQ-32B and DeepSeek-R1 exhibit marginal or negative changes in F1 when using Self-REF (e.g., QwQ-32B: $83.7 \rightarrow 83.0$; DeepSeek-R1: $83.5 \rightarrow 81.1$). This suggests that these models have already internalized process reasoning through reinforcement objectives, and additional weakly supervised signals from self-generated traces may introduce noise rather than improve fidelity. Interestingly, the distilled variant R1-Distill-Qwen-32B, which inherits process judgement behaviors through distillation rather than direct RL training, does benefit modestly from

Table 3: Evaluation results of several models on PROCESSBENCH with solutions as references, including Qwen2.5-32B-Instruct, DeepSeek-R1-Distill-Qwen-32B, QwQ-32B, DeepSeek-R1.

| Model | GSM8K | | | MATH | | | OlympiadBench | | | OmniMATH | | | Average |
|---|---|---|---|---|---|---|---|---|---|---|---|---|---|
| | Error | Correct | F1 | Error | Correct | F1 | Error | Correct | F1 | Error | Correct | F1 | |
| Qwen2.5-1.5B-Instruct | 23.2 | 17.6 | 20.0 | 18.9 | 27.6 | 22.4 | 12.4 | 24.5 | 16.5 | 12.3 | 24.9 | 16.4 | 18.8 |
| w/ Self-REF | 17.4 | 10.4 | 13.0 | 14.0 | 19.7 | 16.4 | 7.4 | 33.6 | 12.1 | 9.0 | 27.8 | 13.6 | 13.8 |
| Qwen2.5-7B-Instruct | 44.9 | 43.0 | 43.9 | 30.1 | 48.0 | 37.0 | 26.5 | 33.3 | 29.5 | 27.1 | 28.2 | 27.7 | 34.5 |
| w/ Self-REF | 45.4 | 82.9 | 58.7 | 26.1 | 80.0 | 39.4 | 15.6 | 72.9 | 25.7 | 15.8 | 70.5 | 25.8 | 37.4 |
| Qwen2.5-32B-Instruct | 43.0 | 97.9 | 59.8 | 33.3 | 95.6 | 49.0 | 22.4 | 90.0 | 35.9 | 22.4 | 87.6 | 35.7 | 45.1 |
| w/ Self-REF | 72.9 | 96.9 | 83.2 | 50.7 | 88.9 | 64.6 | 35.4 | 83.8 | 49.8 | 24.5 | 79.3 | 37.4 | 63.9 |
| R1-Distill-Qwen-32B | 74.4 | 98.4 | 84.7 | 71.5 | 89.4 | 79.5 | 64.3 | 85.5 | 73.4 | 59.0 | 83.8 | 69.3 | 76.7 |
| w/ Self-REF | 77.3 | 96.4 | 85.8 | 76.1 | 92.4 | 83.4 | 68.8 | 90.9 | 78.3 | 58.0 | 87.1 | 69.6 | 79.3 |
| QwQ-32B | 84.1 | 97.4 | 90.2 | 83.0 | 90.4 | 86.5 | 75.9 | 87.3 | 81.2 | 71.1 | 83.8 | 77.0 | 83.7 |
| w/ Self-REF | 82.6 | 93.8 | 87.8 | 81.6 | 88.9 | 85.1 | 77.2 | 85.3 | 81.0 | 71.8 | 83.8 | 77.3 | 83.0 |
| DeepSeek-R1 | 84.1 | 95.3 | 89.3 | 82.3 | 91.1 | 86.5 | 78.2 | 86.4 | 82.1 | 71.7 | 80.9 | 76.0 | 83.5 |
| w/ Self-REF | 81.2 | 93.8 | 87.0 | 83.3 | 87.7 | 85.5 | 74.0 | 79.9 | 76.8 | 70.2 | 79.7 | 74.6 | 81.1 |

Self-REF (F1: 76.7 → 79.3), further supporting that Self-REF is effective in settings where explicit reward-driven reasoning supervision is absent or weak. For the small language models, we can observe that qwen2.5-7B-instruct with Self-REF shows improvements on GSM8K and MATH, while its F1 scores decrease on OlympiadBench and OmniMATH. In contrast, qwen2.5-1.5B-instruct with Self-REF exhibits a decline in performance across all test sets. These results are consistent with our expectations: the effectiveness of Self-REF does depend on the base model's reasoning ability. A weaker instruction-tuned model would generate lower-quality reasoning traces, which would act as noisy or incorrect supervisory signals.

> **Takeaway 3:**
>
> Self-Reference (Self-REF) improves PRM performance primarily for instruction-tuned models lacking RL supervision (e.g., +18.8 F1 for Qwen2.5-32B-Instruct), while RL-trained models (QwQ-32B, DeepSeek-R1) show minimal or negative gains, suggesting limited benefit when process reasoning is already reinforced.

## 4.3 Self-PRM: Can Current PRM Models further Enhance Reasoning Models?

Table 4: Comparison of sampling@k (k = 8, 16, 32, 64) across QwQ-32B and DeepSeek-R1-on three benchmarks under three settings: direct sampling (Pass@), majority voting, BoN with External PRM (BoN w/ PRM), and BoN with Self-PRM.

| Method | @k | QwQ-32B | | | DeepSeek-R1 | | |
|---|---|---|---|---|---|---|---|
| | | AIME24 | AIME25 | CNMO24 | AIME24 | AIME25 | CNMO24 |
| Pass | 1 | 73.3 | 66.7 | 77.8 | 80.0 | 63.3 | 77.8 |
| | 8 | 90.0 | 73.3 | 88.9 | 93.3 | 86.7 | 88.9 |
| | 16 | 90.0 | 80.0 | 94.4 | 93.3 | 90.0 | 88.9 |
| | 32 | 93.3 | 86.7 | 94.4 | 93.3 | 90.0 | 88.9 |
| | 64 | 93.3 | 93.3 | 94.4 | 93.3 | 93.3 | 88.9 |
| Majority Voting | 8 | 80.0 | 76.7 | 83.3 | 80.0 | 76.7 | 77.8 |
| | 16 | 83.3 | 76.7 | 83.3 | 83.3 | 76.7 | 83.3 |
| | 32 | 86.7 | 76.7 | 83.3 | 86.7 | 76.7 | 83.3 |
| | 64 | 86.7 | 76.7 | 83.3 | 86.7 | 76.7 | 83.3 |
| BoN w/ PRM | 8 | 83.3 | 73.3 | 76.7 | 83.3 | 76.7 | 77.8 |
| | 16 | 83.3 | 73.3 | 83.3 | 83.3 | 76.7 | 77.8 |
| | 32 | 86.7 | 76.7 | 83.3 | 86.7 | 76.7 | 77.8 |
| | 64 | 86.7 | 76.7 | 83.3 | 86.7 | 76.7 | 83.3 |
| BoN w/ Self-PRM | 8 | 83.3 | 76.7 | 83.3 | 83.3 | 76.7 | 77.8 |
| | 16 | 86.7 | **80.0** | **88.9** | 86.7 | 76.7 | 83.3 |
| | 32 | **90.0** | **80.0** | **88.9** | 86.7 | **83.3** | 83.3 |
| | 64 | **90.0** | **80.0** | 83.3 | 86.7 | **83.3** | 83.3 |

To evaluate whether current state-of-the-art PRMs can enhance the performance of reasoning models in complex mathematical reasoning tasks, we conduct experiments on AIME24, AIME25, and CNMO24, using Qwen2.5-Math-PRM-72B as the PRM for reranking. Specifically, we apply the Best-of-N with PRM (BoN w/ PRM) strategy, which selects the highest-scoring solution based on the PRM's minimum-step reward among the candidates. As shown in Table 4, models that already exhibit strong reasoning abilities through RL, such as QwQ-32B and DeepSeek-R1, do not benefit from external PRM-based reranking compared to majority voting. Across all datasets and sampling sizes $k = 8, 16, 32, 64$, BoN w/ PRM achieves performance that is comparable to or slightly worse than majority voting. For example, on AIME25 with QwQ-32B, the accuracy plateaus at 76.7 for both BoN w/ PRM and majority voting across all values of k. Similarly, DeepSeek-R1 gains no measurable improvement from PRM reranking on any benchmark. These findings highlight a key limitation of current PRMs: while they are effective in guiding weaker models, they do not effectively enhance or complement already-aligned reasoning models, and may even misalign with their internal reward signals.

Since reasoning models demonstrate strong PRM performance, we introduce a **Self-PRM** method, where each model (e.g., QwQ-32B or DeepSeek-R1) serves as its own PRM to rerank its sampled outputs. As shown in Table 4, the BoN with Self-PRM strategy consistently outperforms both Pass@k and Majority Voting, particularly at larger sample sizes(e.g., $k = 32, 64$). For instance, QwQ-32B on AIME24 improves from 86.7 (majority voting) to 90.0 (BoN w/ Self-PRM) at $k = 32$ and maintains this gain at $k = 64$. Similar improvements are observed across other datasets and for DeepSeek-R1. These improvements suggest that the model's internal reward signal is better aligned with its reasoning behavior than those of externally trained PRMs, enabling a more effective utilization of its latent reasoning capabilities. Thus, Self-PRM offers a more introspective and model-consistent evaluation strategy for complex reasoning tasks.

> **Takeaway 4:**
>
> External PRMs provide little to no benefit for RL-based reasoning models like QwQ-32B and DeepSeek-R1, and may even perform worse than majority voting. In contrast, Self-PRM, where models use their own internal scoring as process reward to rerank outputs, consistently improves accuracy, particularly at higher sampling sizes. This approach demonstrates better alignment with the model's problem-solving abilities in complex reasoning tasks and greater effectiveness in leveraging its latent PRM capabilities.

## 4.4 The Limitations of Self-PRM

Table 5: Model performance comparison across different datasets and problem indices.

| Model | Dataset | AIME24 | | | | AIME25 I | | | | | | AIME25 II | | | CNMO24 | | |
|---|---|---|---|---|---|---|---|---|---|---|---|---|---|---|---|---|---|
| | index | 62 | 63 | 81 | 89 | 6 | 9 | 10 | 12 | 13 | 14 | 4 | 12 | 14 | 1 | 7 | 17 |
| **QwQ-32B** | Difficulty | – | 0/64 | 0/64 | 3/64 | – | 3/64 | – | 4/64 | 2/64 | 0/64 | – | 1/64 | 0/64 | 3/64 | 2/64 | 0/64 |
| | $S_{PRM}$ | – | 10 | 14 | 18 | – | 25 | – | 6 | 2 | 8 | – | 5 | 2 | 55 | 13 | 16 |
| | $S_{TP}$ | – | 0 | 0 | 1 | – | 1 | – | 0 | 0 | 0 | – | 0 | 0 | 2 | 2 | 0 |
| **DeepSeek-R1** | Difficulty | 1/64 | 0/64 | 0/64 | 10/64 | 22/64 | – | 14/64 | 0/64 | 1/64 | 0/64 | 34/64 | – | 6/64 | 19/64 | 0/64 | 0/64 |
| | $S_{PRM}$ | 16 | 5 | 1 | 32 | 44 | – | 11 | 7 | 1 | 1 | 2 | – | 2 | 39 | 7 | 34 |
| | $S_{TP}$ | 0 | 0 | 0 | 3 | 18 | – | 4 | 0 | 0 | 0 | 1 | – | 1 | 10 | 0 | 0 |

*Note*: AIME24 problem indices are sourced from the `math-ai/aime24` dataset on Hugging Face.

The indices in Table 5 were specifically chosen from problems that the respective models (QwQ-32B and DeepSeek-R1) initially failed to solve correctly (i.e., failed at pass@1). This selection allows us to analyze the behavior of Self-PRM on what are considered 'hard problems' for each model. For these hard problems, we sampled 64 solutions to see how many the Self-PRM would judge as correct ($S_{PRM}$), and out of those, how many were truly correct ($S_{PRM}$). The "-" symbol indicates that the problem was not a hard problem for that particular model (i.e., the model solved it correctly on its first try). For instance, in AIME24 index 62, QwQ-32B succeeded initially, so it was excluded from this failure analysis and marked with "-".

To better understand the limitations of Self-PRM, we conduct a fine-grained analysis at the problem level using $N = 64$ sampled solutions per problem instance. For each case, we define $S_{\text{PRM}}$ as the number of solutions labeled as correct by the Self-PRM, and $S_{\text{TP}} \subseteq S_{\text{PRM}}$ as the subset that are truly correct. Thus, the ratio $S_{\text{TP}}/S_{\text{PRM}}$ reflects the precision of Self-PRM's judgment. As shown in Table 5, while Self-PRM often selects a substantial number of candidate solutions as correct (e.g., 55 on CNMO24-index-1 for QwQ-32B), the precision is often low. For example, on CNMO24-index-1, only 2 of the 55 selected solutions are truly correct, yielding a precision of 3.6%. This trend is especially pronounced for high-difficulty problems, such as CNMO24 and certain AIME25 instances, where Self-PRM misclassifies a large number of incorrect reasoning traces as valid. In contrast, DeepSeek-R1, while selecting fewer candidates overall, tends to show higher precision in Self-PRM filtering. On the same CNMO24-index-1 problem, DeepSeek-R1 selects 39 solutions as correct, 10 of which are true positives, resulting in a much higher precision of 25.6%. This suggests that stronger solvers exhibit more reliable self-PRM behaviors, possibly due to more coherent internal alignment between their reasoning and scoring mechanisms. These observations highlight a key challenge: although Self-PRM improves performance at the aggregate level, its fine-grained precision can vary substantially, especially on difficult tasks, where the model tends to overestimate the correctness of its own outputs. Future work may focus on strengthening a model's Self-PRM capabilities through joint training with fine-grained process supervision or continued RL scaling, enabling tighter alignment between the model's problem-solving ability and PRM capabilty.

> **Takeaway 5:**
>
> Self-PRM also suffers from low precision on hard problems, misclassifying many incorrect solutions as correct. Stronger models like DeepSeek-R1 exhibit more reliable self-evaluation. Improving Self-PRM may require joint training with PRM-related tasks or continued RL scaling to enhance internal alignment.

# 5 Conclusion

This work challenges the prevailing assumption that explicit process-level supervision is necessary for enabling effective reasoning in large language models. Through systematic empirical analysis, we demonstrate that RL alone – without access to fine-grained reasoning annotations – can endow models with strong PRM capabilities. RL-trained models such as DeepSeek-R1 and QwQ-32B not only solve complex problems with high accuracy but also implicitly learn to distinguish between valid and flawed reasoning steps. Furthermore, we find that existing state-of-the-art PRMs fail to enhance the performance of slow-thinking reasoning models, and in some cases, underperform compared to simple baselines like majority voting. In contrast, leveraging internal signals, i.e., Self-PRM, consistently improves performance, particularly at larger sampling sizes. These findings reveal a tight coupling between problem-solving and process-level judgement in RL-trained models. Overall, our findings suggest that PRM may not be essential for enhancing complex reasoning, as pure RL not only improves problem-solving skills but also inherently fosters robust PRM capabilities. We hope these findings provide actionable insights for building more reliable and self-aware complex reasoning models.

# 6 Limitations

In this work, since mathematical problems offer clear, objective correctness criteria and structured, step-by-step solutions, we chose the math reasoning domain for studying the co-evolution of problem-solving and process supervision capabilities. Our primary goal was to first systematically establish and validate the existence of the strong positive correlation between a model's problem-solving accuracy and its process judgment F1 score, a phenomenon we demonstrate persists across multiple datasets as RL training progresses. However, a deeper investigation of the underlying mechanism is a valuable direction.

## Acknowledgments

This work was supported by National Key Research and Development Program of China under Grant 2022YFA1004102, and in part by National Natural Science Foundation of China under Grant 62502192, and National Natural Science Foundation of China under Grant 62272210.

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
