# OpenReview forum: "Is PRM Necessary? Problem-Solving RL Implicitly Induces PRM Capability in LLMs"
_NeurIPS.cc/2025/Conference — NeurIPS 2025 poster_

### Official Review · Reviewer_kCye · 2025-06-05

**Clarity:** 2
**Significance:** 3
**Originality:** 3
**Rating:** 3
**Confidence:** 4

**Summary:**

This paper challenges the conventional necessity of Process Reward Models (PRMs) in enhancing the reasoning capabilities of large language models (LLMs). Through empirical analysis, the authors demonstrate that reinforcement learning (RL) training alone, without explicit PRM supervision, can implicitly develop robust PRM capabilities in models like DeepSeek-R1 and QwQ-32B. These RL-trained models not only excel in problem-solving but also exhibit strong process-level judgment, outperforming models explicitly fine-tuned with PRM-labeled data. The study introduces "Self-PRM," a framework where models autonomously evaluate and rerank their solutions using internal rewards. However, Self-PRM struggles with low precision on challenging problems, often misclassifying incorrect solutions. The findings suggest that PRMs may not be essential for complex reasoning, as RL training inherently fosters both problem-solving and process-supervision abilities.

**Questions:**

Besides the weaknesses above, further questions are as follows:

- Self-REF is only evaluated using one non-reasoning model (Qwen2.5-32B-Instruct). If we use a weaker model, such as Qwen2.5-7B-Instruct or Qwen2.5-1.5B-Instruct, will the performance of Self-REF be worse? I am wondering if the performance of Self-REF relies on the reasoning abilties of the model since the improvement of Qwen2.5-32B-Instruct w/Self-REF is more significant on easy tasks (GSM8K) than difficult tasks (Omni-MATH).
- How does the ProcessBench performance evaluated in Figure 1? Do the authors prompt the trained models to answer the first errorneous step in the reasoning process?
- How does Self-PRM rerank the output? Is it the same way as the evaluation method of ProcessBench?
- How do the authors divide the reasoning step in BoN w/ PRM experiments?
- In Table 5, how do the authors choose these problem indices and why some places are filled with "-"?
- Will the code be open-sourced? It would be helpful for the community to reproduce the results and further explore the findings.

**Ethical Concerns:**

["NO or VERY MINOR ethics concerns only"]

**Final Justification:**

I keep my score since my concerns are not fully addressed. The explainations are detailed here (https://openreview.net/forum?id=ooiHIklvN5&noteId=yfKd8Nlz4O).

**Limitations:**

This paper does not discuss limitations.

**Paper Formatting Concerns:**

No.

**Quality:**

2

**Strengths And Weaknesses:**

#### Strengths

- The idea is novel and interesting.
- This paper provides insightful analysis on the relationship between RL, problem-solving abilities, and process supervision capabitlities. Recent works on LLM reasoning and RL rarely focus on the point of PRMs, but this paper provides valuable and interesting insights.
- The paper is easy to follow.

#### Weaknesses

- The improvement of Self-PRM is not significant compared with majority voting and BoN w/ PRM.
- This paper mainly focuses on mathematical reasoning tasks, which may limit the generality of the findings. It would be beneficial to explore the implications of the findings in other domains, such as coding.
- This paper does not discuss limitations.

Minor:
- The Justification should be filled in the checklist.
- L232: SSelf-PRM -> Self-PRM.

I am willing to adjust the rating if the authors can address the weaknesses and the questions below.

---

> ### Author Rebuttal · Authors · 2025-07-31
>
> We sincerely thank the reviewer for their detailed feedback and for recognizing the novelty of our idea and the value of our analysis. We are grateful for your constructive criticism and the opportunity to clarify our work. We will address all weaknesses and questions, and we have a clear plan to revise our paper accordingly.
>
> Q1: On the significance of Self-PRM's improvement
>
> We acknowledge that the improvements in Table 4 might seem modest at first glance. However, we would like to highlight three key points:
>
> 1. **Consistency:** Self-PRM consistently matches or outperforms both majority voting and the external PRM across nearly all benchmarks and sampling sizes. In contrast, the external BoN w/ PRM is sometimes worse than simple majority voting, indicating that its guidance can be misaligned.
>
> 2. **Meaningful Gains on Hard Problems:** On challenging benchmarks like AIME, even a small percentage increase represents solving additional difficult problems. For example, with QwQ-32B on AIME24 (@k=32), Self-PRM achieves 90.0\% accuracy, a notable improvement over both baselines at 86.7\%.
>
> 3. **Computational Efficiency:** Crucially, Self-PRM achieves these gains with **virtually zero additional computational overhead**, as it uses internal reward signals calculated during the initial generation pass . An external PRM, however, can nearly double the inference cost. When factoring in efficiency, the benefit of Self-PRM becomes highly significant. We will add a discussion on this cost-benefit analysis to the paper.
>
> Q2: On the generality of findings (mathematical reasoning)
>
> This is a valid point. We chose this domain because mathematical problems offer clear, objective correctness criteria and structured, step-by-step solutions, making it an ideal "laboratory" for studying the co-evolution of problem-solving and process supervision capabilities.
>
> Currently, there are no process reward evaluation benchmarks similar to ProcessBench in other domains, and constructing such benchmarks is challenging. Therefore, we have not extended the analyses in Table 2 and Table 3 to other domains. Regarding the results in Table 4, we have extended our investigation to the GPQA-diamond dataset, which contains a total of 198 questions specifically designed for the most challenging, PhD-level science questions. The specific results are presented below and we  found that the conclusions remain consistent.
>
> | QwQ-32B           | k=8   | k=16  | k=32  | k=64  |
> |-------------------|-------|-------|-------|-------|
> | Pass@k            | 81.3  | 85.9 | 88.9  | 89.9
> | Majority Voting@k | 64.1 | 63.6  | 65.2  | 64.1 |
> | BoN w/ PRM@k      | 64.6 | 63.6 | 65.2 | 64.6 |
> | Self-PRM@k        | 62.6 | **64.3**| **65.7**| **66.2**|
>
>
> | DeepSeek-R1        | k=8   | k=16  | k=32  | k=64  |
> |--------------------|-------|-------|-------|-------|
> | pass@k         | 93.4  | 93.9  | 94.9  | 95.6  |
> | Majority Voting@k | 82.8  | 82.8  | 82.8  | 82.3  |
> | BoN w/ PRM@k     | 81.8  | 82.8  | 83.3  | 82.3  |
> | Self-PRM@k    | 80.8  | 81.3  | 82.6  | **83.8**  |
>
> Regarding generalization, we hypothesize that the core finding will indeed transfer to other complex reasoning domains like coding, planning, or legal analysis. The underlying principle—that an agent learning to solve a complex task (the policy) simultaneously develops an internal model for evaluating the quality of its actions (the value function)—is a fundamental concept in reinforcement learning. As long as a domain requires multi-step, logical reasoning, we expect a similar synergistic development. We will explicitly add this discussion to our limitations section and highlight cross-domain validation as a critical direction for future work.
>
>
> Q3: On the lack of a limitations section
>
> We sincerely apologize for this oversight. We will add a dedicated "Limitations" section in the revised manuscript. This section will address the domain generality issue mentioned above and other points to provide a more balanced perspective on our work.
>
> Q4: Self-REF on weaker models
>
> In the Self-REF section, we have supplemented the results of qwen2.5-7B-instruct and qwen2.5-1.5B-instruct.
> Based on the results, we can observe that qwen2.5-7B-instruct with Self-REF shows improvements on GSM8K and MATH, while its F1 scores decrease on OlympiadBench and OmniMATH. In contrast, qwen2.5-1.5B-instruct with Self-REF exhibits a decline in performance across all test sets. These results are consistent with our expectations: the effectiveness of Self-REF *does* depend on the base model's reasoning ability. A weaker instruction-tuned model would generate lower-quality reasoning traces, which would act as noisy or incorrect supervisory signals. We will add this analysis to the final paper.
>
> | **Model**               | **GSM8K**               |              |              | **MATH**                |              |              | **OlympiadBench**       |              |              | **OmniMATH**            |              |              | **Average** |
> |-------------------------|-------------------------|--------------|--------------|-------------------------|--------------|--------------|-------------------------|--------------|--------------|-------------------------|--------------|--------------|-------------|
> |                         | **Error**               | **Correct**  | **F1**       | **Error**               | **Correct**  | **F1**       | **Error**               | **Correct**  | **F1**       | **Error**               | **Correct**  | **F1**       |             |
> | qwen2.5-7B-instruct  | 44.9                    | 43.0         | 43.9         | 30.1                    | 48.0         | 37.0         | 26.5                    | 33.3         | 29.5         | 27.1                    | 28.2         | 27.7         | 34.5        |
> | w/ Self-REF             | 45.4                    | 82.9         | 58.7         | 26.1                    | 80.0         | 39.4         | 15.6                    | 72.9         | 25.7         | 15.8                    | 70.5         | 25.8         | 37.4        |
> |                         |                         |              |              |                         |              |              |                         |              |              |                         |              |              |             |
> | qwen2.5-1.5B-instruct  | 23.2                    | 17.6         | 20.0         | 18.9                    | 27.6         | 22.4         | 12.4                    | 24.5         | 16.5         | 12.3                    | 24.9         | 16.4         | 18.8        |
> | w/ Self-REF             | 17.4                    | 10.4         | 13.0         | 14.0                    | 19.7         | 16.4         | 7.4                     | 33.6         | 12.1         | 9.0                     | 27.8         | 13.6         | 13.8        |
>
>
> Q5: ProcessBench evaluation in Figure 1
>
> For the evaluation shown in Figure 1, we prompt the model to act as a "Generative PRM". For each instance in ProcessBench (which contains a problem and a candidate solution), the model is asked to analyze the solution and output a judgment on its correctness, identifying the first erroneous step if one exists. We then compare the model's textual judgment to the ground-truth label to calculate the ProcessBench F1 score.
>
> Q6: Self-PRM reranking method
>
> We adopt the same evaluation method as used in ProcessBench to judge the correctness of each step. The self-PRM evaluates the correctness of all steps in each response, and then among the responses where all steps are correct, we select the final answer based on the consistency of the answers.
>
> Q7: Reasoning step division in BoN w/ PRM
>
> Following the protocol outlined in the ProcessBench paper, we utilize Qwen2.5-72B-Instruct for step segmentation. We will provide the specific prompt in the revised version.
>
> Q8: Problem indices in Table 5: Problem indices in Table 5
>
> The indices in Table 5 were specifically chosen from problems that the respective models (QwQ-32B and DeepSeek-R1) **initially failed to solve correctly** (i.e., failed at pass@1). This selection allows us to analyze the behavior of Self-PRM on what are considered 'hard problems' for each model. For these hard problems, we sampled 64 solutions to see how many the Self-PRM would judge as correct ($S_{PRM}$ ), and out of those, how many were truly correct ($S_{PRM}$ ). The "-" symbol indicates that the problem was **not a hard problem** for that particular model (i.e., the model solved it correctly on its first try). For instance, in AIME24 index 62, QwQ-32B succeeded initially, so it was excluded from this failure analysis and marked with "-". In contrast, DeepSeek-R1 failed the same problem, so its Self-PRM performance is shown. We will clarify this selection methodology in the paper to make the purpose of this fine-grained analysis clear.
> Thank you again for your thorough and helpful review. We are confident that by addressing these points, we can significantly improve the paper and hope you will consider re-evaluating its rating.

---

> > ### Comment · Reviewer_kCye · 2025-08-05
> > **Thanks for your response**
> >
> > Thank you for the detailed response. I still have a few questions and would like to further discuss with the authors, particularly regarding Q1 and Q6.
> >
> > Q1: "can nearly double the inference cost" & Q6
> >
> > Based on the computation method described in Section 2.3 of [1], the inference cost of discriminative PRM should not actually double. Additionally, comparing the wall-clock time of LLM generation and discriminative PRM inference, the PRM inference should be relatively fast, as it only requires decoding a single token theoretically. Also, could the authors provide the exact prompt used for querying the LLM when implementing Self-PRMs?
> >
> > Regarding the claim that Self-PRM incurs "virtually zero additional computational overhead" by leveraging internal reward signals from the initial generation pass, does this mean that Self-PRM is used for reranking outputs that were already generated during that same pass? If so, Self-PRM is not "virtually zero".
> >
> > My concern is that the performance improvement of Self-PRM over Majority Voting or BoN with PRM appears relatively modest. If Self-PRM indeed requires a second generation process, wouldn't that significantly increase the inference cost than Majority Voting or BoN with a discriminative PRM?
> >
> > [1] Large Language Monkeys: Scaling Inference Compute with Repeated Sampling

---

> > > ### Author Response · Authors · 2025-08-07
> > >
> > > Thank you for your insightful questions. We will discuss the concerning from Computational Efficiency and performance.
> > >
> > > Computational Efficiency in Training  vs. Inference Performance Testing
> > > The "Computational Efficiency " claim in our paper refers to the training phase. We demonstrate that a pure Reinforcement Learning (RL) approach can implicitly induce robust Process Reward Model (PRM) capabilities , potentially obviating the expensive data annotation and separate training required for an explicit PRM. In contrast, the comparison in Table 4 is a test of inference-time performance, aimed at evaluating the effectiveness of different reranking mechanisms, not their inference costs.
> > >
> > > Analysis of Inference Efficiency: Majority Voting, Self-PRM, & External PRM
> > > As shown in Table 4, in terms of inference efficiency, Majority Voting is undoubtedly the most efficient strategy. It selects the final answer based solely on the consistency of multiple outputs, requiring no additional model invocations. In contrast, both Self-PRM and External PRM require invoking a large model to evaluate the reasoning process of each candidate solution. The inference efficiency of these two PRM strategies depends primarily on the parameter scale of the model being invoked. In our experiments:
> > > * The External PRM is a 72B model.
> > > * When using QwQ-32B as a Self-PRM, its efficiency is comparatively higher due to its smaller model size.
> > > * Conversely, when using DeepSeek-R1 as a Self-PRM, its large model size would result in significantly slower inference speed compared to the 72B External PRM.
> > >
> > > Therefore, the core of the experiment in Table 4 is not to compare efficiency, but rather to answer a key question: "After incurring a similar or different inference cost, which reranking strategy is most effective at improving the final problem-solving accuracy?" The results show that, despite varying efficiencies, the Self-PRM strategy yields consistent and effective performance gains.
> > >
> > > The prompt we use for Self-PRM is as follows:
> > > > The following is a math problem and a solution (split into paragraphs, enclosed with tags and indexed from 0):
> > > >
> > > > [Math Problem]
> > > >
> > > > {problem}
> > > >
> > > > [Solution]
> > > >
> > > > {tagged_response}
> > > >
> > > > Your task is to review and critique the solution paragraph by paragraph. Once you identify an error in a paragraph, return the index of the paragraph where the earliest error occurs. Otherwise, > > > return the index of -1 (which typically denotes "not found").
> > > >
> > > > Please put your final answer (i.e., the index) in \boxed{{}}.

---

> ### Comment · Reviewer_kCye · 2025-08-09
> **Official Comment by Reviewer kCye**
>
> Thank you for your response. However, PRMs are known to be difficult to generalize across policy models (i.e., models that generate solutions) and tasks [1, 2]. Using Qwen2.5-Math-PRM-72B as the discriminative PRM for comparison is not a fair choice, since this PRM was not trained on the long-CoT responses produced by reasoning models such as QwQ-32B and DeepSeek-R1. As a result, it is out-of-distribution (OOD) for evaluating the responses of these reasoning models. Although this PRM has 72B parameters, its generalization ability remains limited.
>
> Given this OOD issue, Self-PRM still shows only marginal improvement over the discriminative PRM, while incurring a significantly higher inference cost. Regarding inference cost, I suggest the authors compute it using the total inference FLOPs equation provided in Section 2.3 of [3], since model size alone is not an adequate measure. The paragraph-by-paragraph critique process of Self-PRM introduces substantially higher computational cost than that of a discriminative PRM.
>
> For these reasons, I maintain my current score.
>
> References
>
> [1] ProcessBench: Identifying Process Errors in Mathematical Reasoning.
>
> [2] The Lessons of Developing Process Reward Models in Mathematical Reasoning.
>
> [3] Large Language Monkeys: Scaling Inference Compute with Repeated Sampling.

---

### Official Review · Reviewer_GVjb · 2025-07-01

**Clarity:** 3
**Significance:** 2
**Originality:** 2
**Rating:** 4
**Confidence:** 3

**Summary:**

This paper investigates whether explicit process reward models (PRMs) are necessary for strong reasoning in language models. Through thorough experiments, the authors demonstrate that reinforcement learning (RL) can implicitly induce PRM-like abilities in large language models. They further propose “self-PRM,” a method that uses the model itself to judge and rerank its outputs, thereby acting as its own PRM. Self-PRM leads to improved overall performance, though it can exhibit low precision on particularly difficult problems.

**Questions:**

Does models with self-PRM suffer from low precision on difficult problems more than no-PRM? It is not clear from the paper. Maybe I missed it.

**Ethical Concerns:**

["NO or VERY MINOR ethics concerns only"]

**Final Justification:**

Regarding AC's questions:

- Yes, the revised framing does address the overclaiming issue. However, fully shifting the narrative may be nontrivial. For example, Table 3 suggests that Self-PRM is primarily beneficial for models that haven’t been RL-trained, and Section 4.4 acknowledges limitations of self-reflection.
- If the external PRM is not a reasonable baseline (e.g., due to OOD mismatch), then this does weaken the strength of the comparison. That said, I’m not an expert on this specific setup and can’t fully judge its appropriateness.
- I don’t consider math-only scope a major concern.
- I view the paper primarily as an analysis piece, and in that light, it provides valuable insights. For instance, that models can naturally acquire process judgment capabilities during RL. If it were framed purely as a new method, I’d find the evidence less compelling.

**Limitations:**

yes

**Paper Formatting Concerns:**

N.A.

**Quality:**

3

**Strengths And Weaknesses:**

### Strengths
1. The experimental setup is comprehensive, covering a wide range of benchmarks, models, and evaluation methods.
2. I appreciate that the paper is framed more as a careful analysis rather than simply a new method introduction. The authors are transparent about the limitations of their proposed method (self-PRM).

### Weaknesses
I believe there is some misalignment between the paper’s framing and the experimental findings.
1. The results primarily show that external PRMs can be suboptimal for strong RL-trained models, but that self-PRM (using the model itself) is still beneficial. This may reflect the limitations of current external PRMs rather than the broader claim that PRMs are unnecessary. In other words, the experiments demonstrate that current PRMs may be insufficient, but not that process-level guidance itself lacks value.
2. The results indicate that gains in process judgment often precede gains in answer accuracy during RL training, which suggests that guidance via process-level supervision could still be helpful.

---

> ### Author Rebuttal · Authors · 2025-07-31
>
> We sincerely thank the reviewer for their careful reading and insightful feedback. We are very grateful that you appreciated the comprehensive experimental setup and our transparent approach to analyzing the results and limitations of our proposed method. Your main criticism regarding the potential misalignment between our framing and findings is very well-taken, and we would like to clarify our position.
>
> Q1: Regarding the Framing:  "PRMs are Unnecessary'' vs. "Current PRMs are Insufficient''
>
> This is an excellent and sharp observation. We agree that our results do not dismiss the value of process-level guidance itself. In fact, our findings strongly support its value. Our central argument, perhaps not framed as clearly as it could be, is aimed at the conventional approach of building separate, external, and explicitly annotated PRMs. Our title, "Is PRM Necessary?", is meant to question this specific paradigm, which faces major challenges like prohibitive annotation costs and vulnerability to reward hacking.
>
> Our work demonstrates that:
>
> 1. Pure RL can implicitly develop this process-level judgment capability without explicit supervision.
>
> 2. The resulting internal capability can be harnessed via Self-PRM to achieve performance gains, even outperforming existing external PRMs on strong models.
>
> Therefore, we are not arguing that process supervision is without value. Rather, we are proposing that it is possible to achieve effective process supervision *without* a separate, externally trained PRM. Self-PRM is presented as a more direct, scalable, and model-consistent way to leverage this valuable guidance. We will revise the paper's introduction and conclusion to refine our framing, shifting the emphasis from "PRMs are unnecessary" to "A separate, externally-annotated PRM may not be necessary," making this more nuanced argument clearer.
>
> Q2: Regarding Process Judgment Gains Preceding Accuracy Gains
>
> Thank you for highlighting this key result from Figure 1. You are absolutely correct in your interpretation: the fact that process judgment improves early in training *does* suggest that process-level supervision could be a very effective and dense signal to guide RL. We see this not as a contradiction, but as a crucial piece of the puzzle that our work helps illuminate. Our contribution is showing that this powerful early signal does not necessarily need to come from an external PRM. Instead, it is a capability that emerges organically. This opens up exciting future possibilities, such as using Self-PRM *during* RL training to create a self-improvement loop, a point we will emphasize more in our future work section.
>
> Q3: Does Self-PRM suffer from low precision on difficult problems more than no-PRM?
>
> In our experiments, "no-PRM" refers to baselines like majority voting. Majority voting does not produce a confidence score or a precision metric in the same way; it simply selects the most frequent answer. The key finding is that while Self-PRM's precision can be low on difficult problems (i.e., it is overconfident and labels many flawed solutions as valid ), its ranking is still highly effective. As shown in Table 4, using Self-PRM to select the single best candidate (BoN w/ Self-PRM) consistently leads to higher final accuracy than both majority voting and direct sampling. This indicates that although Self-PRM's absolute judgment is imperfect, its relative judgment is superior to the baselines. We will add a paragraph to Section 4.4 to make this important distinction clear.
>
> Thank you once again for your constructive criticism. We believe your feedback will help us significantly sharpen the paper's message and better contextualize our contributions.

---

> > ### Comment · Reviewer_GVjb · 2025-08-06
> >
> > Thank you for the thoughtful and detailed response. I appreciate the clarification that the work is not dismissing process supervision itself, but rather questioning the necessity of a separate, externally trained PRM. The planned reframing and clearer explanation will strengthen the clarity and positioning of the work. I remain positive in my evaluation.

---

### Official Review · Reviewer_shMK · 2025-07-02

**Clarity:** 3
**Significance:** 3
**Originality:** 3
**Rating:** 5
**Confidence:** 4

**Summary:**

This paper investigates whether explicit Process Reward Models (PRMs) are necessary for developing strong reasoning capabilities in large language models, or if pure reinforcement learning (RL) training can implicitly induce these capabilities. The authors conduct a systematic empirical study demonstrating that RL-trained models like DeepSeek-R1 and QwQ-32B develop robust PRM capabilities without explicit process-level supervision, often outperforming models explicitly trained with PRM data. The study reveals a statistically significant correlation between problem-solving proficiency and process supervision capabilities through chi-square analysis. The authors introduce Self-PRM, where models use their internal reward signals to rerank their own outputs, showing consistent improvements over external PRMs and majority voting, particularly at higher sample sizes. However, the analysis reveals limitations in Self-PRM's precision on challenging problems, highlighting the need for better reward alignment.

**Questions:**

1. Generalization across domains (as mentioned in the weakness section): How do these findings extend to non-mathematical reasoning tasks such as coding
2. RL training specifics: The paper shows results for one RL training setup (DAPO with Qwen2.5-7B-Base). How sensitive are the findings to different RL algorithms (PPO, DPO)? Is it possible that different algorithm might exhibit different pattern?
3. Efficiency-wise: What is the computational overhead of Self-PRM compared to external PRMs and majority voting (e.g., wall clock time would be nice to have)? How does this scale with the number of candidate solutions and model size?
4. Threshold: How sensitive is Self-PRM performance to the selection of confidence thresholds or scoring mechanisms? Are there principled ways to set these parameters?

**Ethical Concerns:**

["NO or VERY MINOR ethics concerns only"]

**Final Justification:**

I appreciate that the authors provide results on the additional experiments and show the efficacy of self-PRM outside the MATH domain, as well as answering a couple of questions. All of my concerns have been addressed, and I will modify my score accordingly and vote for acceptance.

**Limitations:**

no.
Justification is "[TODO]" in the paper check list

**Quality:**

3

**Strengths And Weaknesses:**

Strengths:

- This work provides some interesting insights that shed light into the future of PRM
- Comprehensive empirical evaluation: The study covers multiple datasets (GSM8K, MATH, OlympiadBench, OmniMath) and various model types, providing robust evidence for the claims
- I especially like the Chi-square tests provide statistical validation of the correlation between problem-solving and process supervision capabilities across models
- Experiments are well-designed: The RL training experiments with learning curves effectively demonstrate the co-evolution of accuracy and PRM capabilities during training
- Self-PRM offers an alternative to external PRMs with demonstrated usefulness

Weakness:

- The scope of this paper is limited to the math reasoning domain due to the selection of MATH data as the evaluation dataset. It is unclear if the conclusion of this paper will transfer to other domains (e.g., coding)
- Not really a weakness but it'll be interesting to see if self-PRM can be used to further improve the policy its self by acting as a PRM during RL

---

> ### Author Rebuttal · Authors · 2025-07-31
>
> We are extremely grateful to the reviewer for their thoughtful and encouraging feedback, as well as the high scores across the board. We are delighted that you found our work to offer interesting insights, appreciated the comprehensive empirical evaluation and the statistical validation via Chi-square tests, and found the experiments to be well-designed. Your positive assessment is a strong validation of our research direction. We will address the weakness you've pointed out and answer your excellent questions.
>
> Q1: On the Limitation to the Math Reasoning Domain
>
> This is a very fair point, and we agree that the current work's empirical validation is focused on the mathematical domain. We chose this domain because mathematical problems offer clear, objective correctness criteria and structured, step-by-step solutions, making it an ideal "laboratory" for studying the co-evolution of problem-solving and process supervision capabilities.
>
> Currently, there are no process reward evaluation benchmarks similar to ProcessBench in other domains, and constructing such benchmarks is challenging. Therefore, we have not extended the analyses in Table 2 and Table 3 to other domains. Regarding the results in Table 4, we have extended our investigation to the GPQA-diamond dataset (The specific results are presented below.) and found that the conclusions remain consistent.
>
> | QwQ-32B           | k=8   | k=16  | k=32  | k=64  |
> |-------------------|-------|-------|-------|-------|
> | Pass@k            | 81.3  | 85.9 | 88.9  | 89.9
> | Majority Voting@k | 64.1 | 63.6  | 65.2  | 64.1 |
> | BoN w/ PRM@k      | 64.6 | 63.6 | 65.2 | 64.6 |
> | Self-PRM@k        | 62.6 | **64.3**| **65.7**| **66.2**|
>
>
> | DeepSeek-R1        | k=8   | k=16  | k=32  | k=64  |
> |--------------------|-------|-------|-------|-------|
> | pass@k         | 93.4  | 93.9  | 94.9  | 95.6  |
> | Majority Voting@k | 82.8  | 82.8  | 82.8  | 82.3  |
> | BoN w/ PRM@k     | 81.8  | 82.8  | 83.3  | 82.3  |
> | Self-PRM@k    | 80.8  | 81.3  | 82.6  | **83.8**  |
>
> Regarding generalization, we hypothesize that the core finding will indeed transfer to other complex reasoning domains like coding, planning, or legal analysis. The underlying principle—that an agent learning to solve a complex task (the policy) simultaneously develops an internal model for evaluating the quality of its actions (the value function)—is a fundamental concept in reinforcement learning. As long as a domain requires multi-step, logical reasoning, we expect a similar synergistic development. We will explicitly add this discussion to our limitations section and highlight cross-domain validation as a critical direction for future work.
>
> Q2: On Using Self-PRM to Improve the Policy during RL
>
> This is not a weakness but a fantastic suggestion for future work, and we thank you for it. You have accurately identified a natural and exciting extension of our findings. The current work uses Self-PRM in a post-hoc reranking setting. Using the model’s own internal reward signal to provide dense, step-level feedback during RL training could create a powerful self-improvement loop, potentially accelerating learning and pushing performance even further. This is an excellent idea that directly builds on our work, and we will add it to our future work section as a promising research avenue.
>
> Q3: Sensitivity to RL algorithms
>
> Due to time constraints, we directly tested the performance of the open-source model Open-Reasoner-Zero-7B, which is based on the qwen2.5-7B-base model and trained using the PPO algorithm. On the Math, GSM8K, OlympiadBench, and OmniMath datasets, the problem-solving accuracy (%) of Open-Reasoner-Zero-7B are 77.9, 89.8, 42.0, and 37.4 respectively. On ProcessBench, its F1 scores (%) are 41.3, 58.8, 28.0, and 21.1 respectively. Based on these results, we can draw the same conclusion: after RL training, both the problem-solving ability and PRM capability of the model have improved.
> In the revised version, we will retrain a model based on the PPO algorithm, plot the corresponding curves in Figure 1, and provide a more detailed analysis.
>
> Q4: Computational overhead of Self-PRM
>
> Majority voting outperforms both Self-PRM and external PRMs in terms of efficiency. Majority voting merely derives the final answer based on the consistency of results, whereas both Self-PRM and external PRMs require invoking the model to judge the process scores before selecting the answer. In this paper, Self-PRM and external PRMs are utilized in the same manner; the only distinction lies in that external PRMs are derived from independently trained models, while self-PRM does not require additional training. Therefore, the efficiency of Self-PRM and external PRMs depends solely on the model parameters. The external PRMs employed in this paper is a 72B model, while QwQ-Self-PRM, being a 32B model, exhibits superior efficiency. In contrast, R1-Self-PRM, with a size of 671B, demonstrates significantly slower inference speed.
>
> Q5: Self-PRM sensitivity to thresholds
>
> We adopt the LLM-as-judge approach, where the model is directly prompted to judge the correctness of each step. Since this is a generative model, there is no involvement of a confidence threshold in this process.

---

> > ### Comment · Reviewer_shMK · 2025-08-02
> >
> > I appreciate that the authors provide results on the additional experiments and show the efficacy of self-PRM outside the MATH domain, as well as answering a couple of questions. All of my concerns have been addressed, and I will modify my score accordingly and vote for acceptance.

---

### Official Review · Reviewer_SEpk · 2025-07-04

**Clarity:** 3
**Significance:** 2
**Originality:** 3
**Rating:** 4
**Confidence:** 4

**Summary:**

This work analyzes RL algorithms that focus solely on outcome correctness, such as GRPO, which can enhance LLMs' capability of process supervision. By comparing different versions of open-source LLMs and conducting RL training experiments, the author points out that verifying the final result can still endow models with process supervision abilities. This work also highlights a potential correspondence between problem solving ability and process supervision capability.

**Questions:**

Table 3, for the claim "while RL-trained models (QwQ-32B, DeepSeek-R1) show minimal or negative gains, suggesting limited benefit when process reasoning is already reinforced.", another reason could be the QwQ and R1's performance is higher than Qwen2.5-32B-Instruct and R1-Distill-Qwen-32B. Similarly, the advantage of Qwen2.5-32B-Instruct is 18.8, but the advantage of R1-Distill-Qwen-32B is only 2.6. How about testing a smaller RL-trained model?

It's better to provide more statistical results of Sec 4.4. I think drawing a curve of the pass ratio and precision could be very helpful to support the claim "Self-PRM also suffers from low precision on hard problems, misclassifying many incorrect solutions as correct"

**Ethical Concerns:**

["NO or VERY MINOR ethics concerns only"]

**Final Justification:**

Thanks for the author's detailed response and additional results. However, the claim made in this work has a large scope, and the current results are still not fully convincing me, such as the RLHF and RLVR series models have different behaviours. So I keep my score.

**Limitations:**

The statement made in this work is generalizable, but experiments only involve the mathematical domain.

**Quality:**

3

**Strengths And Weaknesses:**

Strengths:
1. The analysis in L141 "Problem-Solving V.S. Judgment" is insightful.
2. This work is well-written and easy to understand.

Weaknesses:
1. L207–209 state: "indicating that Self-REF can inject meaningful structure into process modeling where it is otherwise absent. In contrast, RL-trained models such as QwQ-32B and DeepSeek-R1 exhibit marginal or negative changes in F1 when using Self-REF". However, Qwen2.5-32B-Instruct has undergone an RL training process. While R1-Distill-Qwen-32B has not, the performance gain is already minimal. Therefore, the experimental results do not sufficiently support the stated conclusion.

2. AIME benchmark only have a few questions, and the performance differences reported in Sec 4.3 are not particularly significant. The conclusion that "External PRMs provide little to no benefit for RL-based reasoning models" may not be broadly generalizable.

---

> ### Author Rebuttal · Authors · 2025-07-31
>
> We sincerely thank the reviewer for their valuable time and constructive feedback. We are pleased that you found the paper well-written, easy to follow, and the analysis in the "Problem-Solving V.S. Judgmen'' section insightful. We would like to address the identified weaknesses and questions to further clarify and strengthen our work.
>
> Q1: On the conclusion regarding Self-REF (Table 3)
>
> We appreciate the reviewer's close reading of this section. There seems to be a slight misunderstanding we wish to clarify. The reviewer states that "Qwen2.5-32B-Instruct has undergone an RL training process'', but according to the source technical reports, this model is a purely instruction-tuned model without reinforcement learning. Qwen2.5-32B-Instruct is a non-thinking model. Our paper explicitly categorizes it this way to test the effect of Self-REF in a setting where process reasoning has not been reinforced. The substantial F1 score improvement of +18.8 for this model provides strong evidence for our claim.
>
> Conversely, R1-Distill-Qwen-32B, while not directly trained with RL, was distilled from DeepSeek-R1 and thus inherits its process judgment behaviors. In fact, R1-Distill-Qwen-32B is a thinking model. Therefore, it represents an intermediate case. The fact that its performance gain from Self-REF is minimal (+2.6) further supports our conclusion that as a model's internal reasoning becomes more aligned through RL (or distillation from an RL model), the benefit of weakly supervised signals from its own generated traces diminishes.
>
> Q2: On the generalizability of the AIME benchmark results (Sec 4.3)
>
> This is a fair point. We acknowledge that the AIME benchmarks contain a limited number of problems. Our intention was to use these highly challenging competition-level problems to test the limits of current PRMs in a difficult, out-of-distribution setting.
>
> While the AIME results alone might not be broadly generalizable, they are presented as complementary evidence to our larger-scale findings on Processbench. Across Processbench, our core finding holds: strong RL-trained models like DeepSeek-R1 and QwQ-32B consistently outperform models explicitly trained with PRM labels. The fact that an external PRM also fails to provide a significant boost over majority voting on the difficult AIME problems further reinforces our main thesis. In our revision, we will soften the language in this section to more accurately reflect the limited sample size of the AIME benchmark while maintaining the core insight.
>
> We have supplemented the analysis on the GPQA-diamond dataset, which contains a total of 198 questions specifically designed for the most challenging, PhD-level science questions. The specific results are as follows:
>
> | QwQ-32B           | k=8   | k=16  | k=32  | k=64  |
> |-------------------|-------|-------|-------|-------|
> | Pass@k            | 81.3  | 85.9 | 88.9  | 89.9
> | Majority Voting@k | 64.1 | 63.6  | 65.2  | 64.1 |
> | BoN w/ PRM@k      | 64.6 | 63.6 | 65.2 | 64.6 |
> | Self-PRM@k        | 62.6 | **64.3**| **65.7**| **66.2**|
>
>
> | DeepSeek-R1        | k=8   | k=16  | k=32  | k=64  |
> |--------------------|-------|-------|-------|-------|
> | pass@k         | 93.4  | 93.9  | 94.9  | 95.6  |
> | Majority Voting@k | 82.8  | 82.8  | 82.8  | 82.3  |
> | BoN w/ PRM@k     | 81.8  | 82.8  | 83.3  | 82.3  |
> | Self-PRM@k    | 80.8  | 81.3  | 82.6  | **83.8**  |
>
> Based on the analysis of results on GPQA-diamond dataset, we find that the conclusion remains consistent with our previous findings: External PRMs provide little to no benefit for RL-based reasoning models.
>
> Q3: Testing a smaller RL-trained model (Table 3)
>
> Following your suggestion, we directly used an open-source, smaller RL-trained model: Qwen-2.5-7B-SimpleRL-Zoo.
> The specific results are as follows:
>
>
> | **Model**               | **GSM8K**               |              |              | **MATH**                |              |              | **OlympiadBench**       |              |              | **OmniMATH**            |              |              | **Average** |
> |-------------------------|-------------------------|--------------|--------------|-------------------------|--------------|--------------|-------------------------|--------------|--------------|-------------------------|--------------|--------------|-------------|
> |                         | **Error**               | **Correct**  | **F1**       | **Error**               | **Correct**  | **F1**       | **Error**               | **Correct**  | **F1**       | **Error**               | **Correct**  | **F1**       |             |
> | Qwen-2.5-7B-SimpleRL-Zoo | 42.5                    | 95.3         | 58.8         | 27.4                    | 83.7         | 41.3         | 17.1                    | 76.7         | 28.0         | 12.4                    | 71.0         | 21.1         | 37.3        |
> | w/ Self-REF             | 37.7                    | 93.3         | 53.7         | 26.9                    | 87.7         | 41.2         | 21.3                    | 82.3         | 33.9         | 19.0                    | 78.4         | 30.6         | 39.9        |
>
> Based on the results, we can observe that the conclusions drawn from the 7B model are consistent with our previous findings: RL-trained models show minimal or negative gains, suggesting limited benefit when process reasoning is already reinforced.
>
> Q4: More statistical results for Self-PRM (Sec 4.4)
>
> We agree completely. A visualization would be much more powerful. We thank the reviewer for this constructive idea. In the revised version, we will add a plot showing the curve of the solution pass ratio versus Self-PRM precision. This will provide a clearer and more intuitive illustration to support our claim that Self-PRM's precision is low on difficult problems.
>
> Q5: On the Limitation of the Mathematical Domain
>
> We acknowledge this limitation. We chose the mathematical domain because it offers objective and verifiable ground truth for both final answers and intermediate reasoning steps, making it an ideal environment to systematically study the relationship between problem-solving and process supervision.
>
> Currently, there are no process reward evaluation benchmarks similar to ProcessBench in other domains, and constructing such benchmarks is challenging. Therefore, we have not extended the analyses in Table 2 and Table 3 to other domains. Regarding the results in Table 4, we have extended our investigation to the GPQA-diamond dataset (The results are shown in Q2.) and found that the conclusions remain consistent.
>
> We will explicitly state in the conclusion that our findings are currently confined to mathematical reasoning and that future work should investigate whether these principles generalize to other complex reasoning domains.
>
>
> Thank you again for your insightful feedback. We believe that incorporating these changes will significantly improve the quality and impact of our paper.

---

> > ### Comment · Reviewer_SEpk · 2025-08-04
> >
> > Thanks for the detailed response, I think we have a different understanding of the terminology of RL. Qwen2.5-32B-Instruct is indeed a non-thinking model, but in the Qwen2.5 technical report, Section 4 states that the model has undergone an RL training for improving its truthfulness, helpfulness, conciseness, relevance, harmlessness, and debiasing. This RL process is different from the DeepSeek-R1 and later long cot (thinking) models. Maybe the author should consider using a finer category to describe those reasoning models.

---

> > > ### Author Response · Authors · 2025-08-04
> > >
> > > Your understanding is correct, and we sincerely appreciate your suggestions. Qwen2.5 utilizes RL to align with human feedback, whereas DeepSeek-R1 employs RL to enhance the model's reasoning capabilities. Our primary goal here is to clarify the distinction between thinking models and non-thinking models. In the revised version, we will provide more detailed descriptions of these two types of RL applications and introduce a finer category of RL, aiming to avoid any potential misunderstandings.

---

### Official Review · Reviewer_cPkA · 2025-07-06

**Clarity:** 3
**Significance:** 3
**Originality:** 3
**Rating:** 4
**Confidence:** 3

**Summary:**

Challenging the necessity of external Process Reward Models (PRMs), this paper's systematic investigation reveals that pure reinforcement learning not only improves an LLM's problem-solving skills but also synergistically co-develops its internal process supervision capabilities, suggesting that continued RL scaling is a more direct path to advancing complex reasoning.

**Questions:**

See Weaknesses.

**Ethical Concerns:**

["NO or VERY MINOR ethics concerns only"]

**Final Justification:**

I believe the second part of my concern has been addressed, but the first part remains unclear. Therefore, I'm inclined to keep my score.

**Limitations:**

yes

**Quality:**

3

**Strengths And Weaknesses:**

Strengths
- This paper presents a comprehensive empirical study with a very broad scope. It investigates a range of topics, from the limitations of state-of-the-art Process Reward Models (PRMs) in LLM reasoning to the advantages and disadvantages of a Self-PRM approach. Following the precedent set by Deepseek R1, this work further demonstrates that an external PRM may not be essential for successfully scaling reasoning abilities via reinforcement learning.
- The experimental finding presented in Figure 1 is particularly interesting. It reveals a strong correlation where the improvement in a model's problem-solving ability is accompanied by an enhancement of its process supervision capabilities. This insight could potentially unlock novel applications, such as using the reasoning model itself as a PRM.

Weaknesses
- My main criticism is that the paper attempts to tackle too many broad questions at once. This ambitious scope seems to have come at the cost of depth, leading to some topics being underexplored and a general lack of profound insights.
The paper does not adequately explain why RL scaling enhances the ability of a reasoning model to act as a PRM. Does this process make the internally generated reward signals more robust against being 'hacked' by the policy, or are there other underlying mechanisms at play? I was expecting a dedicated section (e.g., a Section 3.4) with targeted experiments to delve into this crucial question. Given that the main body of the paper is currently under the page limit, there appears to be ample space to include such a vital analysis.
- The study overlooks a critical scenario that merits investigation. A key potential advantage of an external PRM is its ability to provide denser supervision signals during RL training compared to a simple, binary rule-based verifier. However, the paper includes no experiments to test this. If an external PRM can guide the RL process more effectively than a rule-based verifier, it would demonstrate that PRMs are not entirely obsolete and still hold significant value—a conclusion that this paper currently seems to dismiss.

---

> ### Author Rebuttal · Authors · 2025-07-31
>
> We sincerely thank the reviewer for their valuable time and constructive feedback. We would like to address the identified weaknesses and questions to further clarify and strengthen our work.
>
> Q1: Regarding the lack of depth on *why* RL scaling enhances PRM capability
> - Our primary goal was to first systematically establish and validate the *existence* of the strong positive correlation between a model's problem-solving accuracy and its process judgment F1 score, a phenomenon we demonstrate persists across multiple datasets as RL training progresses. However, we agree that a deeper investigation into the underlying mechanism is a valuable addition.
> - In our revision, we will add a new section with targeted analyses to explore this *why* ; these experiments will be conducted after the rebuttal period due to time constraints. Our hypothesis is that as RL training improves the policy, the model's internal value function becomes a more reliable estimator of the correctness of reasoning steps. To test this, we will analyze the evolution of the model's internal reward signals and their correlation with ground-truth step-wise correctness at different checkpoints during training. This will provide a more direct explanation for how RL implicitly fosters robust PRM capabilities without explicit supervision.
>
> Q2: Regarding the overlooked scenario of using an external PRM for RL training
>
> - We thank the reviewer for highlighting this important scenario. We acknowledge that our study did not test the efficacy of an external PRM as a source of dense reward signals during the RL training phase. Our experimental scope was focused on a different, but equally relevant, question: whether existing state-of-the-art PRMs can further enhance already strong reasoning models through post-hoc reranking. Our results show that for models like QwQ-32B and DeepSeek-R1, external PRMs offer no significant benefit over simpler methods like majority voting, whereas an internally derived Self-PRM does.
>
> -  While using a PRM for dense supervision is a valid approach, our work challenges the prevailing assumption that is considered necessary. The paper highlights that conventional PRMs face significant hurdles, including prohibitive annotation costs and vulnerability to reward hacking. Our core finding is that pure RL scaling presents a more direct and efficient path, as it not only boosts problem-solving but also inherently develops the desired process supervision abilities. We believe this finding, which shows that a complex capability can be an emergent property of a simpler training objective, is a significant contribution. We will clarify this distinction in the paper and position the dense supervision angle as a valuable direction for future research. In our subsequent research, we are attempting to use a self-PRM to guide the RL training process, where both the policy model and the self-PRM are improved simultaneously. More related results will be disclosed in future studies.

---

> > ### Comment · Reviewer_cPkA · 2025-08-05
> > **Response to Authors**
> >
> > Thanks for the authors' response. I believe the second part of my concern has been addressed, but the first part remains unclear. Therefore, I'm inclined to keep my score.

---

### Note · Authors · 2025-08-13

To the Chairs, Area Chairs, and Reviewers,

We sincerely thank all reviewers (cPkA, GVjb, kCye, SEpk, shMK) for your invaluable time and profound insights. The peer review process has been an incredibly valuable learning experience, and we will undertake significant revisions based on your feedback.

We are encouraged that the novelty of our core ideas was recognized: that pure RL can implicitly induce PRM capabilities and that problem-solving skills co-evolve with process supervision.

Synthesizing all feedback, we will systematically revise the paper:

1. **Re-framing the Core Argument**: Per Reviewer GVjb's insightful critique, we will pivot from the broad claim that "PRM is unnecessary" to a more precise and defensible thesis: a separate, externally-annotated PRM may not be necessary or optimal for models already strong in reasoning via RL. This better aligns our claims with our evidence.

2. **Addressing Generalizability & Limitations**: Responding to multiple reviewers, we will incorporate our new GPQA-diamond data to bolster generalizability claims and add a dedicated "Limitations" section. This will frankly discuss the need for validation in domains like coding.

3. **Exploring Underlying Mechanisms**: We acknowledge Reviewer cPkA's critical point on the lack of a mechanistic explanation for why RL enhances PRM capability. We will frame this as a key future work direction, outlining plans to probe the model's internal value function.

4. **Nuanced Analysis of Self-PRM**: We will provide a more balanced analysis of Self-PRM. We will directly address the OOD challenge in our external PRM comparison (a key concern from Reviewer kCye), acknowledging the potential for distribution mismatch. We will also transparently differentiate between Self-PRM's training-phase efficiency gains (no annotation needed) and its inference-phase computational costs.

5. **Clarifying Details**: We will refine model classifications (per Reviewer SEpk) to distinguish between RL for alignment and RL for reasoning, and add experimental details to improve reproducibility.

We are confident these revisions will produce a much stronger, more transparent, and valuable paper. We again extend our sincerest thanks for your diligent and constructive contributions.

---

### Decision · Program_Chairs · 2025-09-17

**Decision:**

Accept (poster)

**Comment:**

This paper investigates whether explicit PRMs are necessary for reasoning in LLMs, showing that RL alone can induce process supervision capabilities and introducing Self-PRM as a reranking mechanism. The study is timely, well-executed, and provides valuable empirical insights, though the framing initially overstated claims, Self-PRM’s gains are often modest, and evaluation is limited mostly to math reasoning with potential OOD issues in PRM baselines. Reviewers are split between borderline accept and reject, but agree the revised framing is more accurate. Overall, the work advances understanding of RL vs. PRM in reasoning and is worth sharing.